# Indoor Localization Method for a Mobile Robot Using LiDAR and a Dual AprilTag

**Yuan-Heng Huang and Chin-Te Lin \***

Department of Mechanical Engineering, National Central University, Taoyuan 320317, Taiwan
* Correspondence: chintelin@ncu.edu.tw; Tel.: +886-3-4267379

**Abstract:** Global localization is one of the important issues for mobile robots to achieve indoor navigation. Nowadays, most mobile robots rely on light detection and ranging (LiDAR) and adaptive Monte Carlo localization (AMCL) to realize their localization and navigation. However, the reliability and performance of global localization only using LiDAR are restricted due to the monotonous sensing feature. This study proposes a global localization approach to improve mobile robot global localization using LiDAR and a dual AprilTag. Firstly, the spatial coordinate system constructed with two neighboring AprilTags is applied as the reference basis for global localization. Then, the robot pose can be estimated by generating precise initial particle distribution for AMCL based on the relative tag positions. Finally, in pose tracking, the count and distribution of AMCL particles, evaluating the certainty of localization, is continuously monitored to update the real-time position of the robot. The contributions of this study are listed as follows. (1) Compared to the localization method only using LiDAR, the proposed method can locate the robot's position with a few iterations and less computer power consumption. (2) The failure localization issues due to the many similar indoor features can be solved. (3) The error of the global localization can be limited to an acceptable range compared to the result using a single tag.

**Keywords:** mobile robot; AprilTag; global localization; adaptive Monte Carlo localization (AMCL); LiDAR

## 1. Introduction

For smart factories, the localization and navigation system of mobile robots is the key to implementing automation logistics [1]. The system pre-configuration in the new environment will involve a lot of labor, including creating the map for robot navigation, setting path points, and initializing the system. With the expansion of the factory, these operations will become more complex and costly in terms of time and labor. In addition, the above pre-configuration is mostly regular and repetitive operations. If the pre-configuration operations can be automated, the applicability of automation logistics can be improved.

The mobile robot localization problem can be considered as a kind of robotics coordinate transformation problem. The goal is to synchronize the robot coordinates on the cyber side with the map measured from an actual environment. In addition, using computer vision to establish spatial coordinate relationships is a well-developed technique. Furthermore, several studies [2–6] have applied code tags to positioning problems in many fields, such as gait analysis for medical units, dynamic platform landing for drones, and automatic parking assistance systems. They reported that the code tags are useful and effective in these applications.

The automatic localization methods are broadly classified into five categories, including distance localization, three-point localization, environmental fingerprint localization, tag recognition localization, and map matching localization [7,8]. Each method has its pros and cons, but almost no single category can meet all of the needs of indoor scenarios. As a result, many new methods have developed, combining different sensors and algorithms to approach the needs of increasingly complex scenarios.

Robots based on light detection and ranging (LiDAR) usually use the adaptive Monte Carlo localization algorithm (AMCL) as the localization method. AMCL uses sensor data and known map information to infer the probability distribution of the robot's position on the map and then estimates the robot's position and orientation by optimizing the probability distribution. The AMCL optimization process starts with an initial probability distribution, which can be generated in two ways. The first is to generate a uniform probability distribution over the entire map by assuming that the robot can be located anywhere, in an enumeration-like manner, for subsequent optimization. This approach can automatically locate the robot, but the calculation takes a long time and sometimes induces a kidnapping problem [9]. Another way is to specify the most probable initial position directly through human observation and manual setting, which can significantly shorten the subsequent optimization calculation time. However, with a larger factory size, manual setup procedures' complexity and time cost will grow significantly [10–19].

This study analyzes the main works on solving the localization issues of LiDAR-based robots today, summarizes the localization approaches, and lists the unresolved issues in practical applications

1. Xu et al. [14] used fingerprint localization in a WIFI environment to assist a LiDAR robot in accomplishing global localization. However, due to the limited accuracy, it still takes time to converge AMCL particles to achieve accurate localization. On the other hand, localization is not reliable in factories due to high interference from wireless signals.

2. Some studies use image feature recognition to achieve global localization [12,16,17]. First, the robot is moved around the environment to build a database with the location data and the observed images using vision sensors. When the robot moves, it uses image descriptors or convolutional neural networks (CNN) to solve the robot's pose in the environment by comparing the image similarity. The advantage of this approach is that it does not require any additional map-related configuration. However, due to the tedious database creation process and the high sensitivity to scene features, it is unsuitable for factories with a quick changeover.

3. Other studies [13,15,18,19] have proposed practical solutions using code tags to achieve the global localization of robots. For example, Hu et al. [18] attached the tags to the floor to assist and correct the AMCL positioning. In another study [15], the tags are attached to the room ceiling as the robot's absolute positioning basis and solve the cumulative error problem of the encoder. However, the above methods require many tags placed in the environment, resulting in difficulties in installing and maintaining them.

4. Some works have demonstrated pose estimation with code tags [13,19,20]. The relationship between the tag location and the origin of the grid map can be first recorded. Then, when the robot vision captures the tag, the system can infer the relative relationship between the tag and the vision sensor. Finally, the robot's position on the map can be determined based on forward kinematics. The above methods using fewer code tags can achieve global positioning with high accuracy of 1~2 cm. However, the literature [21] report and the actual tests in this study have observed an unstable variation in the decoded results. The range of variation increased sharply with the distance from the tag, resulting in inaccurate localization. In the literature [9], the error is eliminated by multiple tags, but this method requires additional settings in the environment and has more placement limitations.

This study develops a robotic localization system and proposes an innovative global localization approach to effectively solve both shortcomings of localization using simple LiDAR sensing signals and the defects of the single tag localization method. This study contains five chapters. Section 2 reviews the related works, presents the issues that remain to be solved, and the research objectives. In Section 3, the methodology using a dual AprilTag is explained. Section 4 illustrates the experimental methods and discusses the results. Finally, the conclusions and future research directions are presented in Section 5.

## 2. Related Works

### 2.1. Mobile Robot

Mobile robots use the information collected by sensors to create a digital map and achieve autonomous localization and navigation while moving. The main control processes of a mobile robot in an unfamiliar environment contains several parts. In the process of perception, the robot uses the onboard sensors, such as LiDAR, camera, encoder, and inertial measurement unit, to extract environmental data and convert them into information required for movement. Localization is to identify the robot's position on the digital map according to the available sensing information. Then, the system can plan the robot's possible path with its location and map information. Finally, the robot can decide the movement path through sensing, positioning, and mapping calculations.

### 2.2. Simultaneous Localization and Mapping (SLAM)

The technique to solve the problem of modeling the digital map of a robot in an unknown environment is known collectively as "simultaneous localization and mapping (SLAM)". The reason for the simultaneous implementation of map modeling and localization is that the robot needs to know its location to build a map, but it also needs to obtain its location from the map. Therefore, the robot's movement must be continuously tracked while building the map. In practice, the robot will store the constructed digital map and use it for subsequent navigation, equivalent to directly making the factory map a known reference.

### 2.3. Adaptive Monte Carlo Localization (AMCL)

From a mathematical viewpoint, the robot's localization can be expressed as a state estimation problem, mainly due to the error of the sensor values. Hence, it is necessary to collect and evaluate the information from the environment and the robot and compensate for the bias error using statistical methods. AMCL is today's most popular robot localization algorithm, which adds an adaptive mechanism to particle filter and Monte Carlo localization to properly deal with the robot pose tracking issue.

#### 2.3.1. Particle Filter

The particle filter generates many hypothetical particles representing an arbitrary probability distribution of states. In the absence of any positional information, the possible robot poses can be represented by particles with a uniform distribution; that is, the robot can be in any position. Then, when the robot is moving around, the positional probability of the state can be updated by comparing the sensor data with the digital map.

The state of a group of particles carries can be represented with two parameters:

$$X = \left\{ \langle x^{[j]}, w^{[j]} \rangle \right\}_{j=1,2,\ldots,J} \tag{1}$$

- State hypothesis $x^{[j]}$: the physical meaning and state represented by each particle j.
- Importance weight $w^{[j]}$: the probability that the state represented by the particle j is true.

After normalizing the weights w of all particles, the probability of the discrete particles can be approximated to a continuous function by the kernel density estimation (KDE) method. The form is expressed as:

$$p(X) = \sum_{j=1}^{J} w^{[j]} \delta_{x^{[j]}}(X) \tag{2}$$

where $\delta_{x^{[j]}}(X)$ is the Gaussian distribution with the mean value of x for particle j. By reassigning the original particle weight and resampling the particle distribution according to the new weight value, the proposed function can be effectively modified to a new target function for resampling. Therefore, the particle filter is a recursive Bayesian filter

that continues the iterative prediction and correction process. It is also a non-parametric approach. To summarize the nature of the particle filter:

- Modeling: Using discrete particles to simulate an arbitrary continuous distribution;
- Prediction: Using known information to predict the next state of particle distribution;
- Correction: Corrects the particle distribution by calculating the weight of the particles.

The fitting needs continuous approximation functions determined based on the discrete particles. As a result, the number of particles will affect the performance of the approximation model.

### 2.3.2. Monte Carlo Localization

Monte Carlo localization (MCL) is the application of the particle filter to the robot localization problem. The physical implications of applying all the participating variables of the particle filter to robot localization are listed:

- State hypothesis: The robot pose, including position and orientation.
- Importance weight: The probability that the hypothetical state represents the truth of the particle.
- Proposed particle distribution: The distribution of the proposed particles of the robot in the map and usually provided by sensed features or the previous distribution.
- Target particle distribution: The new particle distribution obtained after updating the proposed particle weights.

In continuous positioning, i.e., iterative particle filtering, the proposed particle distribution is used as a reference for motion control. The proposed particle positions can be defined by the previous state $x_{t-1}$ and the motion command $u_t$:

$$x_t^{[j]} \sim p(x_t \mid\mid x_{t-1,u_t}) \tag{3}$$

The weight used to correct the particle distribution, in the case of a robot using LiDAR for localization, is determined by observing the surrounding information $z_t$. The mathematical equation is described as follows:

$$w_t^{[j]} \propto p(z_t \mid\mid x_{t,m}) \tag{4}$$

The estimation optimization process of AMCL can be divided into initialization, observation, measurement, weight updating, and resampling. Initialization is based on a specified method to generate an initial particle distribution. The observation step is to collect environmental information in situ or after moving. The difference between the environmental information and the digital map is calculated in the measurement step, and the particle weights are updated accordingly. Finally, in the resampling step, the new particle distribution is generated based on the distribution of the updated weights.

### 2.3.3. Short Review of AMCL

To solve the robot positioning problem, AMCL is a common approach. It provides the following features.

(1) ProvidING a possible solution to the robot kidnapping problem: In robot localization, if the robot misestimates its actual position, such as a temporary failure of the LiDAR or loss of signal caused by environmental noise, the traditional MCL cannot detect and solve the kidnapping problem. Therefore, AMCL adds a mechanism by monitoring the average weight of all particles in time and re-scattering some particles globally when the average weight value drops.

(2) Solving the problem of a fixed number of particles: AMCL can detect the distribution dispersion in real-time and change the total number of particles required, thus reducing redundant iterations.

Although the AMCL method using LiDAR solves most of the robot positioning problems, the method still has the following three drawbacks:

(1) Positioning failure with similar features in geometry: The sensing signal of the LiDAR is composed of the distance from the obstacle and the reflection strength. Therefore, if there are many similar spaces in a field, it is difficult for the robot to recognize its position.

(2) Limitation to solving the robot kidnapping problem: The mechanism of AMCL to solve the kidnapping problem is to randomly scatter the points in space again, so a certain chance of failure will exist. However, this approach does not guarantee to solve the kidnapping problem anytime.

(3) With the increased power consumption and convergence time of the map size: As the factory map becomes large, more particles are needed for initialization to ensure coverage. Because of the probability calculation by points, the number of particles on the map significantly affects the performance of AMCL global localization.

### 2.4. AprilTag

AprilTag [22,23] is a code tag positioning technology that captures a specific 2D tag through the camera. After calculation, the coordinate transformation of the tag relative to the camera can be extracted for further localization or applications. The system has excellent features and is widely used in various applications, such as augmented reality (AR), robot localization, and camera calibration. The following is a brief description of the system's key features:

- Low decoding complexity: AprilTag decoding is simpler and more efficient than QRcode coding. It meets the real-time requirements.
- High reliability: Compared to other 2D codes, the AprilTag algorithm can automatically detect and locate tags over longer distances, with lower fractional variation, uneven illumination, large rotations, or background clutter [22].
- Multi-tag detection: The system can simultaneously detect and locate multiple tags on a single shot.

AprilTag can be used to find the coordinates of tags in space, including rotation and translation, and was therefore applied to solve robot positioning problems in this study.

### 2.5. Gaps and Research Objective

Suppose mobile robots can quickly perform pose estimation and global localization, especially find the working locations of production equipment autonomously. In that case, this will help to improve the flexibility of the automation logistics system to respond to changes. However, the past works did not consider the partnership between mobile robots and production equipment. This study proposes the tag-based localization method by attaching two AprilTags to the front side of production equipment. Once the positions of the equipment are changed, the mobile robots can still move to the correct positions for loading or interaction.

## 3. Methodology

This chapter first describes the robot localization system designed using LiDAR and a dual AprilTag. Then, the innovative approach to the global localization problem is detailed. Finally, after analyzing the effects of localized errors using the single tag localization method, the advantages using a dual AprilTag to eliminate errors are explained.

### 3.1. Robotic Positioning System Process

The procedure of the localization system in this study is shown in Figure 1. First, the global localization problem is solved by finding the initial position through the code tags method proposed in this study when the robot starts, as shown in Figure 2a. Then, the initial pose is used to determine the particle distribution required for AMCL to perform pose tracking. Next, the particle weights are continuously updated by comparing the

odometry data and LiDAR information with the map profile while the robot is moving. After that, the particle distribution is updated with the new weights to realize the pose tracking, as in Figure 2b. Finally, during the positional tracking, the resampled particle distribution data are analyzed, and the variance of the distribution is used to determine whether a kidnapping problem happens. If the variance exceeds a specific domain value, the system will perform the repositioning based on the code tags again, as in Figure 2c. In this process, one of the contributions of this study is the use of dual tag vision to assist global localization.

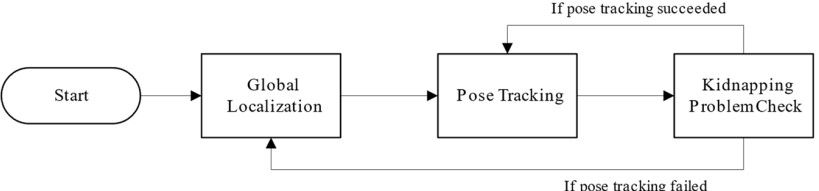

**Figure 1.** The procedure of the localization system in this study.

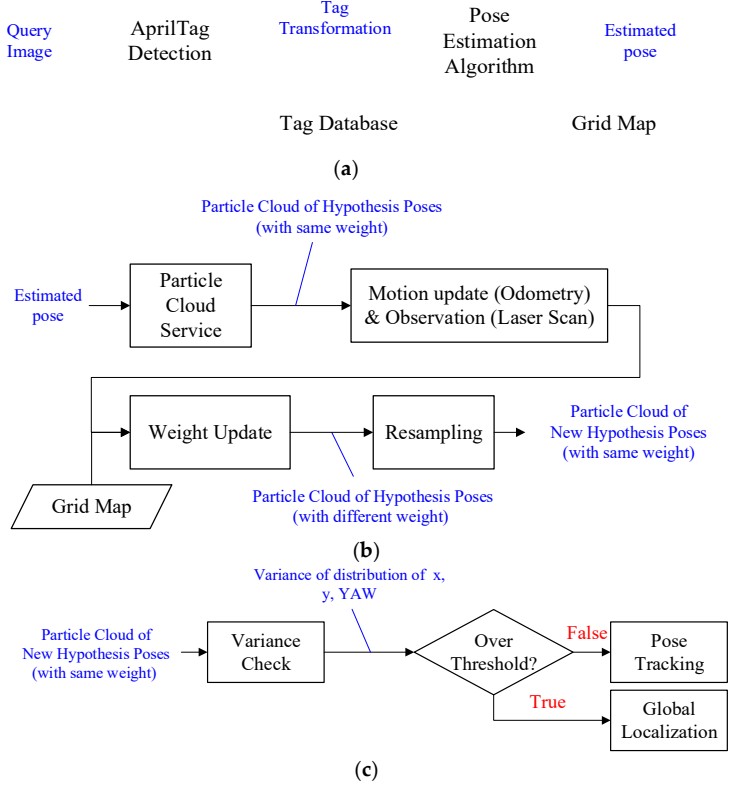

**Figure 2.** Detail process in the localization system of this study. (**a**) Global localization; (**b**) pose tracking; (**c**) kidnapping problem check.

### 3.2. Pose Estimation Using a Dual AprilTag

Figure 3 is a schematic diagram of the proposed pose estimation concept. Figure 3a shows the situation before global localization. In this situation, the digital map is block A, while the robot observes the environment, marked as contour B. Since the localization is not yet successful, the two are not aligned. The proposed pose estimation tries to align the two entities so that the digital map can act as a digital twin of the physical field and provide navigation information to the mobile robot. Hence, a reference mechanism is needed to regulate the degrees of freedom between the two. If only one common reference point is given, there will exist a rotation between the two, which cannot guarantee their alignment, as shown in Figure 3b. The innovative approach proposed in this study is to generate two

common reference points through a pair of neighboring tags, called dual AprilTags, which provides sufficient restrictions to estimate robot pose, as shown in Figure 3c. Another alignment approach is based on the position and orientation of the reference point, using a single AprilTag, but the uncertainty of decoding may cause errors. Therefore, the paired neighboring tags attached to equipment are suitable to assist pose estimation and guide the boarding of mobile robots.

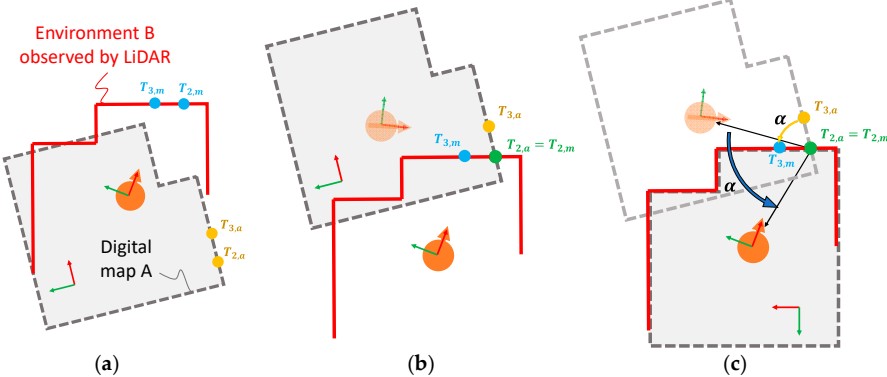

**Figure 3.** Schematic diagram of the concept of pose estimation: (**a**) miss alignment; (**b**) location correction with one reference point; (**c**) alignment with two reference points.

The robot pose estimating problem using the dual AprilTag in a factory map can be simplified as shown in Figure 4. $O_M$ and $(X_M$ and $Y_M)$ are the original point and coordinates of the factory map. There are other positions in the factory map, including AprilTags $T_2$ and $T_3$ and Robot $R$. If the robot is not localized well, the value of the robot position is incorrect and is denoted as $R_m$, and the AprilTags will be observed in positions of $T_{2,m}$ and $T_{3,m}$. Here, both triangles of $\Delta T_{3,m} O_{Ri} T_{2,m}$ and $\Delta T_3 O_R T_2$ are congruent triangles. The objective of the simplified problem is to find the robot's exact position $R$.

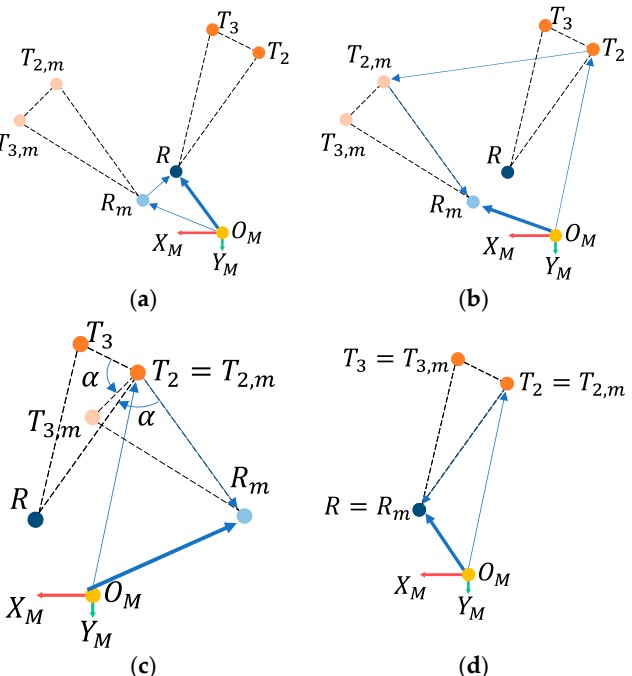

**Figure 4.** Approaches to determine robot position using the dual AprilTag; (**a**) localization problem; (**b**) shift; (**c**) rotate; (**d**) result.

The relationship between the evaluated and true position of the robot is shown in Figure 4a and can be expressed as:

$$\overrightarrow{O_M R} = \overrightarrow{O_M R_m} + \overrightarrow{R_m R} \tag{5}$$

Pose estimation can be achieved after eliminating the error denoted by $\overrightarrow{R_m R}$, which cannot be achieved by using simple LiDAR information directly. Therefore, AprilTag provides geometric correlations for reference. As the camera in the robot observes the AprilTags, AprilTag technology will provide the geometric relationship from the camera to the tags. Since the actual locations of AprilTags, $T_2$ and $T_3$, are ready on the grid map, the first reference point can be one of the tags, for example, using $T_2$. Then, the evaluation position of the robot $\overrightarrow{O_M R_m}^0$ observed through the reference point is shown in Figure 4b and can be expressed as:

$$\overrightarrow{O_M R_m}^0 = \overrightarrow{O_M T_2} + \overrightarrow{T_2 T_{2,m}} + \overrightarrow{T_{2,m} R_m} \tag{6}$$

By subtracting $\overrightarrow{T_2 T_{2,m}}$ in both sides of the equation, i.e., letting $T_{2,m}$ coincide with $T_2$, as shown in Figure 4c, the first evaluation position of the robot can be obtained:

$$\overrightarrow{O_M R_m}^1 = \overrightarrow{O_M R_m}^0 - \overrightarrow{T_2 T_{2,m}} = \overrightarrow{O_M T_2} + \overrightarrow{T_{2,m} R_m} \tag{7}$$

The observation error of the rotation causes the $\overrightarrow{T_{2,m} R_m}$ to be inconsistent with the actual $\overrightarrow{T_{2,m} R}$. Here, another tag, e.g., $T_3$, is used to eliminate the rotational error in the observation. Let $\hat{t}$ and $\hat{t}_m$ be the unit vectors of $\overrightarrow{T_2 T_3}$ and $\overrightarrow{T_{2,m} T_{3,m}}$, respectively, and the angle between the two vectors is $\alpha$. Then, their relationship is:

$$\hat{t}_m = \boldsymbol{R}(\alpha)\,\hat{t} \tag{8}$$

where the rotation matrix is:

$$\boldsymbol{R}(\alpha) = \begin{bmatrix} \cos\alpha & -\sin\alpha \\ \sin\alpha & \cos\alpha \end{bmatrix} \tag{9}$$

and the value of $\alpha$ is:

$$\alpha = \cos^{-1}\left(\hat{t}_m \cdot \hat{t}\right) \tag{10}$$

Then, the relationship between $\overrightarrow{T_{2,m} R_m}$ and $\overrightarrow{T_2 R}$ is:

$$\overrightarrow{T_2 R} = \boldsymbol{R}(-\alpha) \cdot \Delta \overrightarrow{T_{2,m} R_m} \tag{11}$$

Therefore, the final evaluation pose of the robot is shown in Figure 4d and can be expressed as:

$$\overrightarrow{O_M R_m}^2 = \overrightarrow{O_M T_2} + \boldsymbol{R}(-\alpha) \cdot \Delta \overrightarrow{T_{2,m} R_m} = \overrightarrow{O_M T_2} + \overrightarrow{T_2 R} = \overrightarrow{O_M R} \tag{12}$$

After the above steps, the robot's position and orientation on the map can be found by the dual AprilTag.

### 3.3. Performance Analysis Using AprilTag

Next, Figure 5 illustrates the performance of AprilTag coordinate transformation. The coordinates transformation $T_{O_M}^{O_{T,a}}$ from the map system to an AprilTag can be expressed as:

$$T_{O_M}^{O_{T,a}} = \begin{bmatrix} \cos\theta_{M,Ta} & -\sin\theta_{M,Ta} & x_{M,Ta} \\ \sin\theta_{M,Ta} & \cos\theta_{M,Ta} & y_{M,Ta} \\ 0 & 0 & 1 \end{bmatrix} \tag{13}$$

Here, $x_{M,Ta}$, $y_{M,Ta}$, and $\theta_{M,Ta}$ are the relative distances and angle from the map system to the AprilTag. The coordinate transformation from the measured tag coordinates system to the initial robot coordinates system $T_{O_{T,m}}^{O_{Ri}}$ is:

$$T_{O_{T,m}}^{O_{Ri}} = \begin{bmatrix} \cos\theta_{Tm,Ri} & -\sin\theta_{Tm,Ri} & x_{Tm,Ri} \\ \sin\theta_{Tm,Ri} & \cos\theta_{Tm,Ri} & y_{Tm,Ri} \\ 0 & 0 & 1 \end{bmatrix} \tag{14}$$

Here, $x_{Tm,Ri}$, $y_{Tm,Ri}$, and $\theta_{Tm,Ri}$ are the relative distances and angle from the tag to the robot. Therefore, the transformation using the single AprilTag can be expressed as:

$$T_{O_M}^{O_R} = T_{O_M}^{O_{T,a}} T_{O_{T,m}}^{O_{Ri}} \tag{15}$$

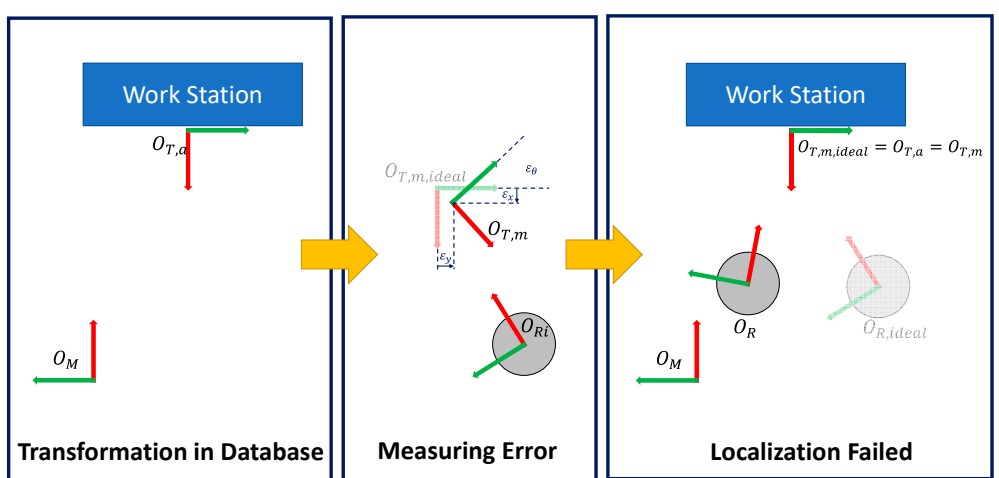

**Figure 5.** Schematic diagram of the inaccuracy of the single tag global positioning algorithm due to observation error.

This equation indicates that the coordinate transformation of the measured tag for the map origin is derived from the coordinate transformations from the map origin to the robot and from the robot to the tag. When inferring robot positioning, the coordinates conversion from map origin to the tag and the coordinates conversion from the tag to a robot is used. If this relationship is completely correct, Equation (15) can lead to the correct spatial localization result.

However, there are usually translation errors $(\varepsilon_x$ and $\varepsilon_y)$ and rotation errors $\varepsilon_\theta$ between the measured tag $O_{T,m}$ and the ideal tag $O_{T,m,ideal}$. Therefore, Equation (14) can be modified to:

$$T_{O_{T,m}}^{O_{Ri}} = \begin{bmatrix} \cos(\theta_{Tm,Ri}+\varepsilon_\theta) & -\sin(\theta_{Tm,Ri}+\varepsilon_\theta) & x_{Tm,Ri}+\varepsilon_x+r(1-\cos\varepsilon_\theta) \\ \sin(\theta_{Tm,Ri}+\varepsilon_\theta) & \cos(\theta_{Tm,Ri}+\varepsilon_\theta) & y_{Tm,Ri}+\varepsilon_y+r\sin\varepsilon_\theta \\ 0 & 0 & 1 \end{bmatrix} \tag{16}$$

where the actual distance between the tag and the robot is $r'$ and the ideal distance is $r$. The ideal and actual positioning results of the robot are $(x, y, \theta)$ and $(x', y', \theta')$, respectively. The relationship between the actual and theoretical positioning results of the robot is:

$$\Delta O_R = \begin{bmatrix} \Delta x_R \\ \Delta y_R \\ \Delta \theta_R \end{bmatrix} = \begin{bmatrix} x' - x \\ y' - y \\ \theta' - \theta \end{bmatrix} = \begin{bmatrix} (\varepsilon_x + r'(1 - \cos\varepsilon_\theta))\Delta\cos\theta_{M,Ta} \\ -(\varepsilon_y + r'^{\sin\varepsilon_\theta})\Delta\sin\theta_{M,Ta} \\ \varepsilon_\theta \end{bmatrix} \tag{17}$$

In Equation (17), as the distance between the tag and the robot becomes large, the rotational error has more influence on the positioning result. The relationships are plotted in Figure 6.

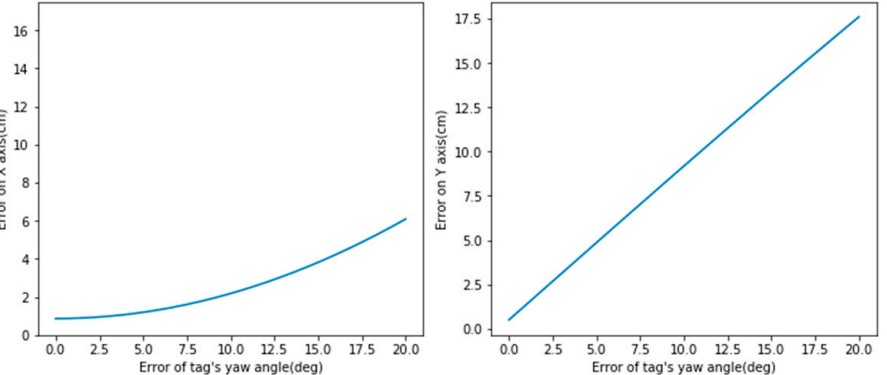

**Figure 6.** The relationship between the observed distance and translation and rotation error under $\varepsilon_x = \varepsilon_y = 1, k = 100, \theta_{M,Ta} = 30°$.

With the above derivation, the robot localization method with a single code tag is unreliable enough for practical applications. Therefore, this study proposed the dual AprilTag approach to solve the problems. Furthermore, the rotation parameter of the tags is not used in Equation (12), so the interference of the observed rotation error can be avoided.

## 4. Experiments and Results

This chapter describes the experimental procedure and method to verify the theory in Section 3, which is divided into the following parts: concept validation, global localization performance evaluation, and single and two tag global localization accuracy and stability study.

### 4.1. Experimental Devices

This study used FESTO's Cyber-Physical Factory (CP Factory) as the main experimental environment. The mobile robot used is FESTO Robotino, equipped with SICK-S300 LiDAR and Logitech C920 Pro camera (Figure 7). Please refer to Tables 1 and 2 for the specifications. The robot operating system (ROS) middleware is built on the hardware of Robotino to control the motion, execute algorithms and collect sensing information.

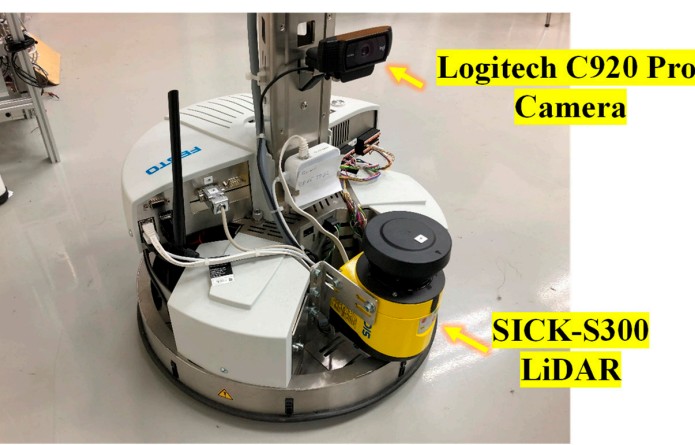

**Figure 7.** FESTO Robotino with LiDAR and camera.

**Table 1.** Specification of SICK-S3000S300.

| Target | Specification |
| --- | --- |
| Scan angle | 270° |
| Protective field range | 2 m |
| Reflectivity | Reflectors 1.8% . . . > 1000% |
| Response time | 80 ms |
| Resolution | 30 mm, 40 mm, 50 mm, 70 mm, selectable |
| Angular resolution | 0.5° |
| Protective field supplement | 100 mm |
| Warning field range | 8 m (at 30% reflectivity) |
| Distance measuring range | 30 m |
| Number of multiple samplings | 2 . . . 16, configurable vis CDS |
| Reset time | 2 s . . . 60 s, configurable |

**Table 2.** Specification of Logitech C920 Pro.

| Target | Specification |
| --- | --- |
| Max resolution | 1080 p/30 fps–720 p/30 fps |
| Focus type | Autofocus |
| Lens type | Glass |
| Built-in mic | Stereo |
| Diagonal field of view(dFoV) | 78° |

*4.2. Prof of Concept*

The feasibility of the proposed method in the real environment was verified. First, a digital map and a tag database of the factory were constructed, as shown in Figure 8. Then the global localization algorithm of this study was executed at several random locations to observe the localization results.

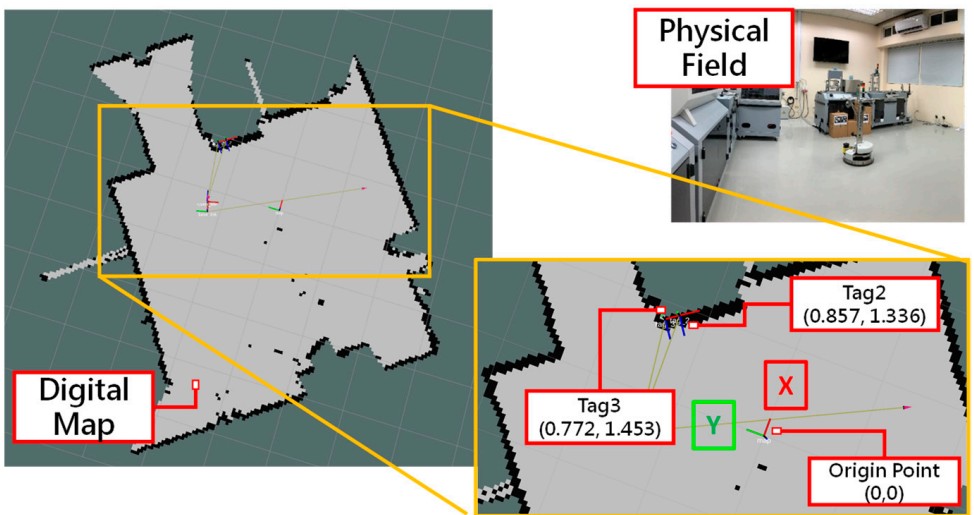

**Figure 8.** Digital map and tag database.

For when the robotic camera captures two neighboring tags, the before/after particle distribution using the proposed method is shown in Figure 9. This result shows the convergence of the particle distribution and proves the method's feasibility. In addition, this method can also be applied to more than two tags, as shown in Figure 10. The algorithm will pair the tags separately and then find the optimal solution by the least square method.

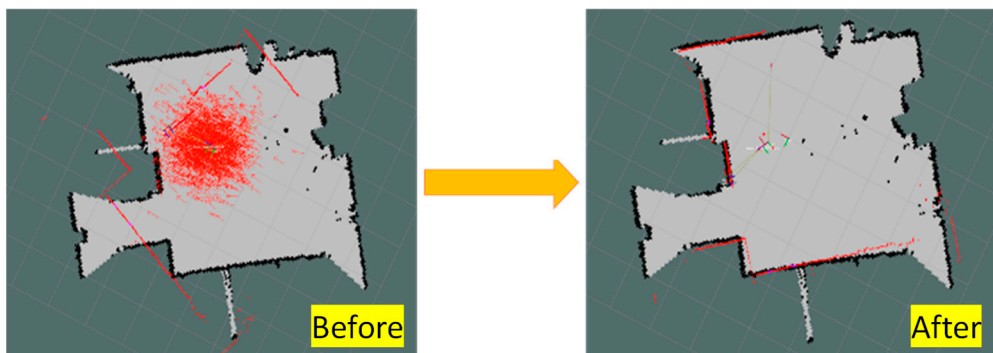

**Figure 9.** Results of global localization using the dual AprilTag.

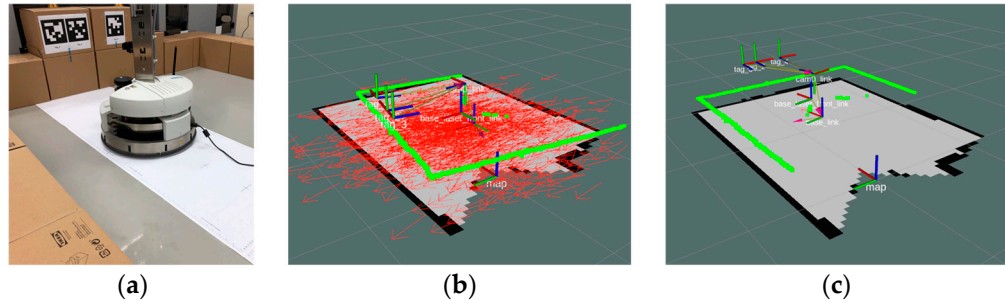

**Figure 10.** Results of global localization using a multi AprilTag; (**a**) a mobile robot and three AprilTags; (**b**) before localization; (**c**) result.

### 4.3. Global Localization

In this experiment of dynamic global positioning, the robot moves and locates simultaneously using two methods. The first is the classic method that performs AMCL only using LiDAR sensing information. The other is the proposed method in this study, performing AMCL using LiDAR and a computer vision approach with the dual AprilTag. The test fields consist of a factory case with significant differences in features and a trap case with similar features in geometry, as shown in Figures 11a and 12a, respectively. Furthermore, the robot's paths are shown in Figures 11b and 12b, and the particle states of AMCL are recorded at specific locations in the path points. Finally, the performance difference between the two methods is compared in terms of the variance of particle distribution and the total number of particles.

#### 4.3.1. Factory Case

In the factory case, the particle distributions at different path points are shown in Figure 13, when the classic and the proposed methods are implemented. The result using the classic method shows that a huge number of the particles must be distributed over the map at the beginning for the robot to determine its position. Then, the particles are converged by AMCL until the positioning is completed. In contrast, the proposed method allows the robot to obtain a precise position at the beginning with a limited number of particles on the map. Observing the particle variance of the two methods in the X and Y directions in Figure 14, the proposed method can achieve global localization directly at the beginning and omit the particle convergence steps of the classic method.

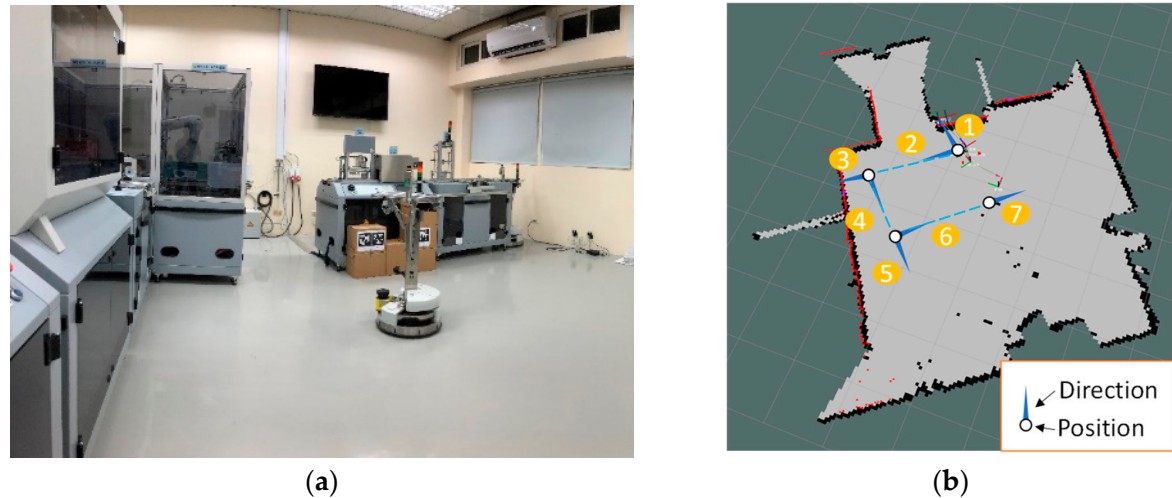

(**a**)                                                                     (**b**)

**Figure 11.** Factory case field; (**a**) a factory; (**b**) robot moving path.

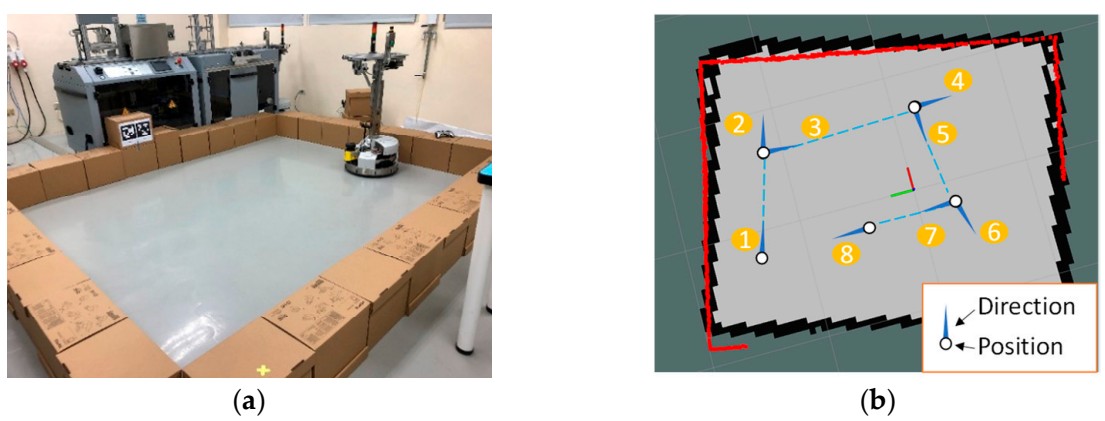

(**a**)                                                                     (**b**)

**Figure 12.** Trap case field; (**a**) a designed rectangle bound using boxes; (**b**) robot moving path.

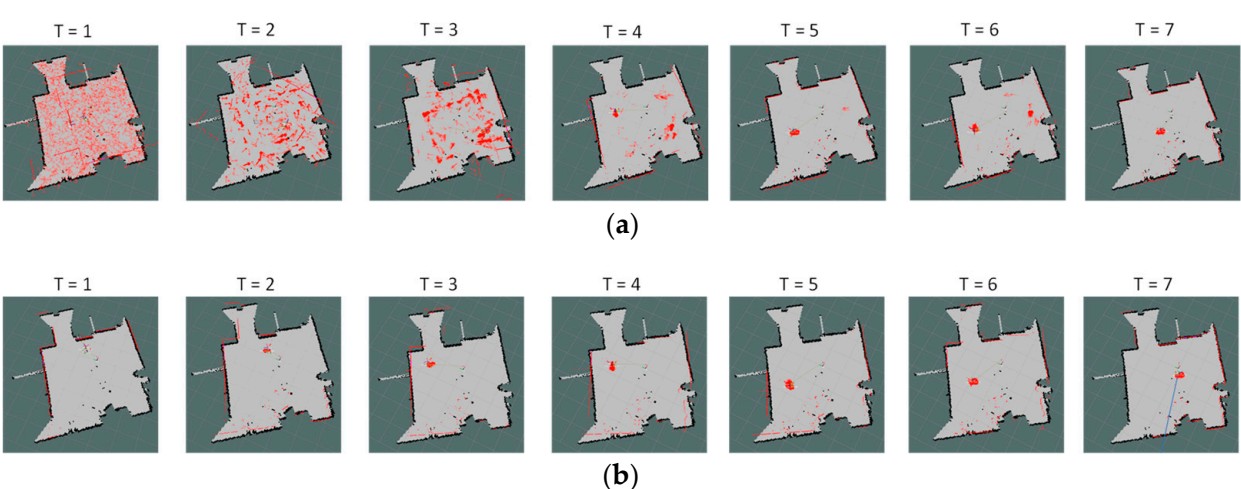

(**a**)

(**b**)

**Figure 13.** Particle distributions in path points during global localization in the factory case study.
(**a**) Classic; (**b**) proposed.

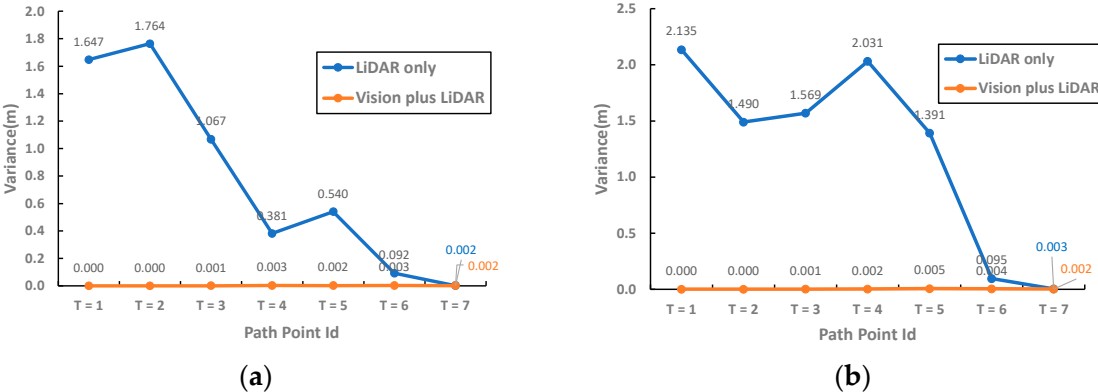

**Figure 14.** Convergence of particle state in path points during global localization in the factory case study. Variance of particles on (**a**) the *X*-axis and (**b**) the *Y*-axis.

On the other hand, since the classic method needs to exhaust all the locations on the map, many particles are required in the initial stage to ensure that the entire map is covered. The larger the map area is, the more particles are required, which means the required computing power is high and the computation time is long. In contrast, the initial conditions are given by a dual AprilTag in the proposed method, so that the initial number of particles used in the AMCL algorithm is not related to the map area. Figure 15 shows the number of particles of the two methods at each path point. Compared with the global localization by simple LiDAR, the proposed method requires only a small number of particles to complete the localization and subsequent pose tracking, and the performance is almost the same as that of the manual assignment. Thus, the computing power the proposed method requires to generate and evaluate particles is less than what the classic method needs.

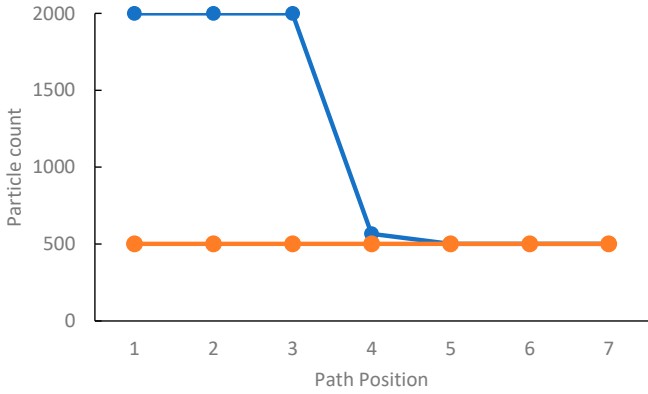

**Figure 15.** Particle used during global localization in the factory case (the setting range is 500~2000).

### 4.3.2. Trap Case

The particle distribution recorded at the eight path points in the trap case using the classic method is shown in Figure 16a. It is observed that the particles are still not fully converged after several iterations using AMCL. According to the variation of particle distribution in the X and Y directions shown in Figure 17, the particles do not converge definitely. Furthermore, similar edge profiles are observed in the two dense particle-swarm at the positions of T > 4, as shown in Figure 16a, resulting in the difficulty for the robot to localize with high similarity of geometric features. In contrast, the particle distribution using the proposed method is initially close to convergence, as in Figure 16b. Benefiting from the suitable initial conditions, the robot can quickly and effectively perform positional tracking and localization in environments with high geometric similarity.

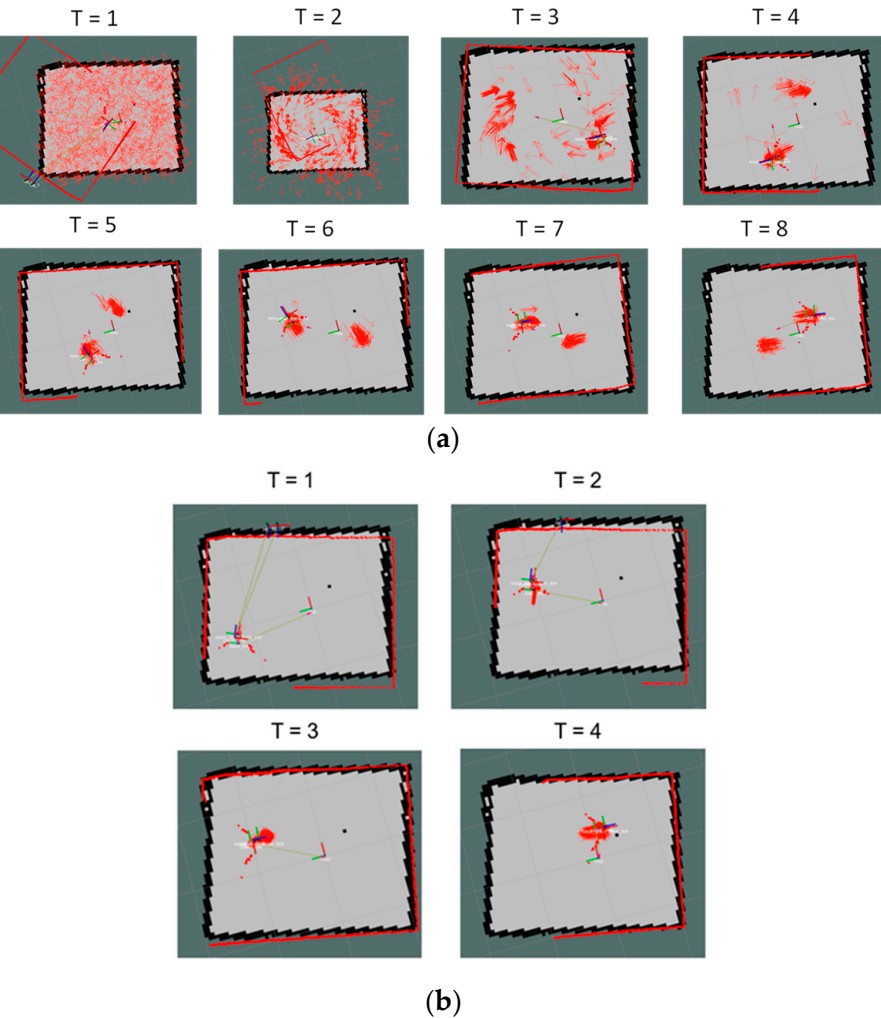

**Figure 16.** Particle distributions in path points during global localization in the trap case study. (**a**) Classic; (**b**) proposed.

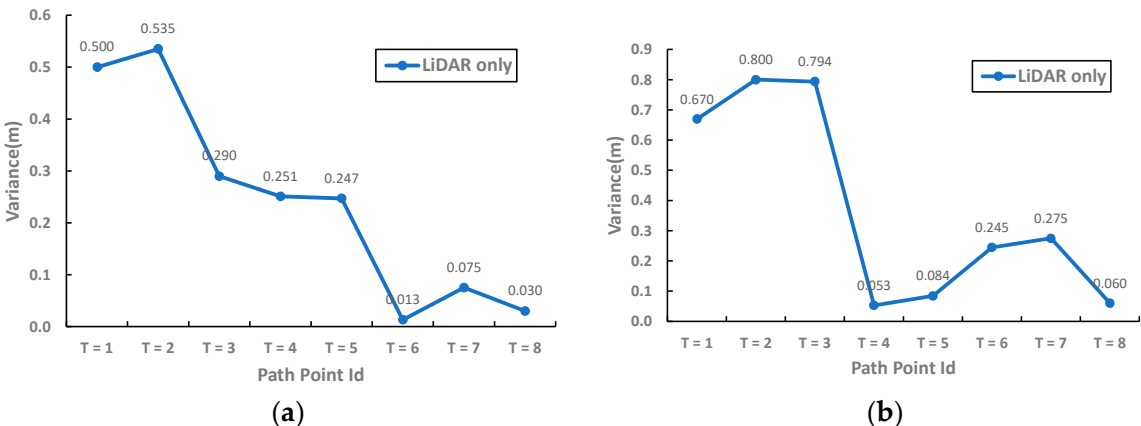

**Figure 17.** Convergence of particle state in path points during global localization in the trap case study. Particle variance on (**a**) tje *X*-axis and (**b**) the *Y*-axis.

### 4.4. Stability of a Single Tage and Two Tags

This experiment evaluates the stability of the global localizations using a single tag and a dual tag. The latter is proposed in this study. First, the overall experimental environment is schematically set up, as shown in Figure 18. Then, the global localization algorithm of the two methods is executed at five different distances. Finally, the previous step is repeated several times, and the measured results are recorded.

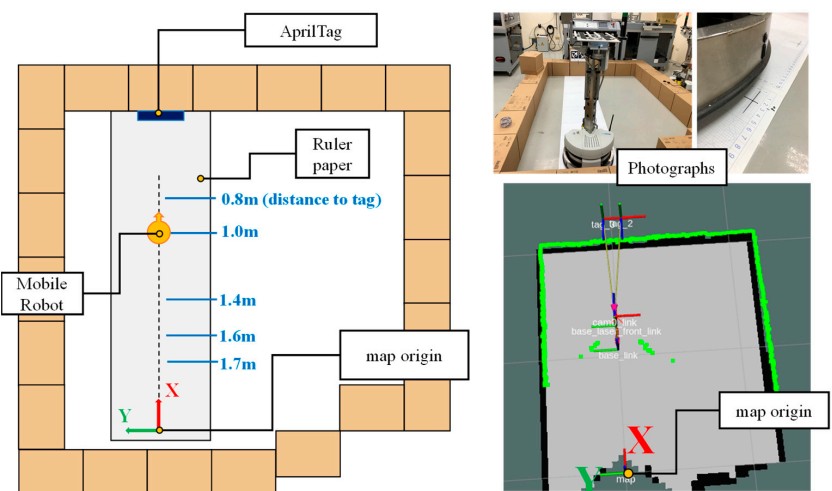

**Figure 18.** Experimental environment for the stability of a single tag and two tags.

Figure 19 shows the error situations of the results after repeatedly executing the global localizations at five positions. The horizontal axis indicates the distance between the robot and the tags, and the vertical axis indicates the error amount. The accuracy and stability of the single tag method decrease as the robot is far away from the tag. Figure 19b,d shows that the positioning error using a single tag is highly correlated with the rotation error of the tag. However, the proposed dual tag method can maintain a small error variation. Therefore, it can be confirmed that the proposed dual tag method is better than a single tag approach. On the other hand, it is possible for a mobile robot to observe several tags simultaneously. The mobile robot can process the tags, pair by pair, and use statistical methods to accurately identify its position.

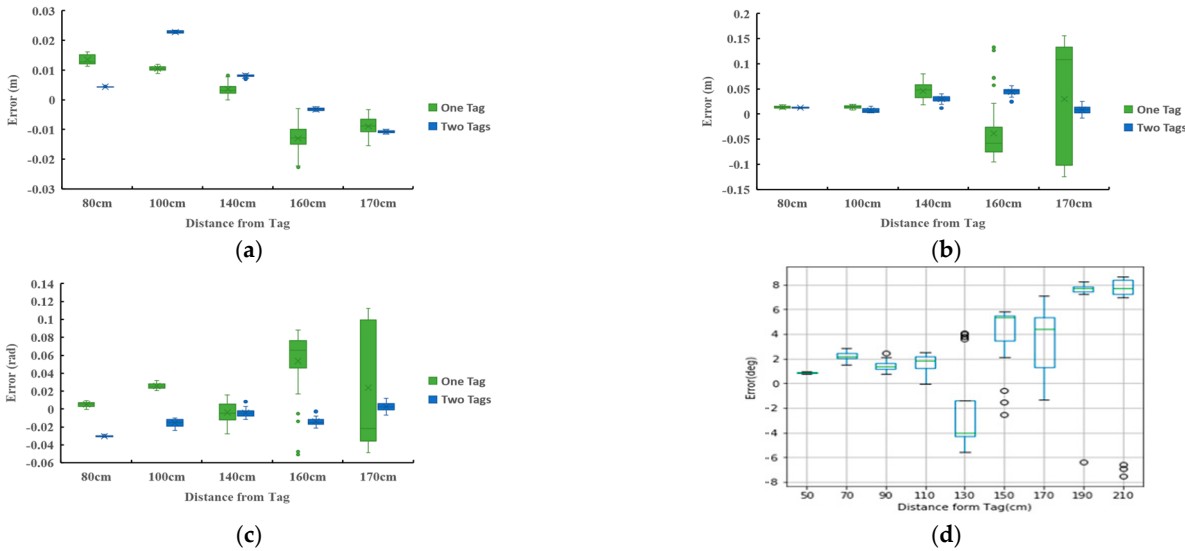

**Figure 19.** The relationships between localization error and relative distance. (**a**) Error on the x direction; (**b**) error on the y direction; (**c**) error on rotation; (**d**) rotation error from the AprilTag.

## 5. Conclusions

This study designs an integrated vision and LIDAR ROBOT localization method using two AprilTag tags for the robot pose estimation, which can effectively solve the factory's global localization and kidnapping problem. After practice and experiment, the following contributions were obtained from this study:

1. The method proposed in this study can complete mobile robot localization as long as the camera captures more than two AprilTag tags. This feature is also helpful in solving the kidnapping problem caused by multiple similar features.
2. Compared with the classical method, only using LiDAR and AMCL, the proposed method can effectively reduce the particle distribution variability and the number of particles used in AMCL. This feature can enhance the efficiency of pose estimation and improve the performance of global localization.
3. This study avoids using AprilTag rotation information with variation errors and successfully prevents its impact. The translation and rotation errors in the test environment are within the acceptable range. The proposed method is helpful for the precise mobile robot localization.

Besides logistics in factories, the proposed method using two AprilTags can potentially apply in other applications with trap issues to mobile robots, especially for a workspace within many similar scenes, such as large warehousing, rows of horticultural plantations, hospital automation, and restaurant services. The proposed method allows mobile robots to quickly estimate their positions to an object through the two neighbor tags on the object and solve the trap issues.

Although this study establishes various testing scenarios, the reliability and practicality of the proposed method for industrial application are still worthy of further investigation. Two suggestions are listed for future works: (1) to investigate the causes of the defective AprilTag identification tags and to solve the fundamental tag identification problem; and (2) to use faster response cameras to deal with dynamic positioning issues due to the image quality in dynamic shots.

**Author Contributions:** Conceptualization, Y.-H.H. and C.-T.L.; methodology, Y.-H.H.; software, Y.-H.H.; validation, Y.-H.H. and C.-T.L.; resources, C.-T.L.; data curation, Y.-H.H.; writing—original draft preparation, Y.-H.H.; writing—review and editing, C.-T.L.; visualization, Y.-H.H. and C.-T.L.; supervision, C.-T.L.; project administration, C.-T.L. All authors have read and agreed to the published version of the manuscript.

**Funding:** This research was funded by National Science and Technology Council grant number [MOST 111-2218-E-008-007].

**Data Availability Statement:** No new data were created or analyzed in this study. Data sharing is not applicable to this article.

**Conflicts of Interest:** The authors declare no conflict of interest.

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
