# Peer review of "Indoor Localization Method for a Mobile Robot Using LiDAR and a Dual AprilTag"

_electronics, doi:10.3390/electronics12041023_

Round 1

Reviewer 1 Report

The approach is interesting, and the presentation of the theoretical aspects are very good.

The mathematical model is well founded and adequate to the methods presented.

Author Response

Thank you for your review and for recognizing our research.

Reviewer 2 Report

The paper is well presented with a appropriate level of details, explaining the problem, proposition of the solution and the results.

The main issue I would like to indicate is a not convincing explanation of the need of the dual-tag approach in section 3.2. The authors claim that knowing a position of a single tag it is not enough to estimate the pose, because there may be arbitrary orientation error (authors do not mention it limit, so the suggestion is it may be +- pi from the real value, an example in Fig 3b shows the error exceeding pi/2). It is not correct because:

1) tags are one sided, so if a tag is observed we already have a limited orientation error +- pi/2;

2) practical detection is in even smaller angular range - various research indicate practical angle from the tag axis in which it was still detected as around +-pi/4

3) finally AprilTag detection functions return tag rotation matrix in camera frame (although the estimate can have  quite a high error)

As the result, the estimate from a single tag is not expected to be as bad as the authors present (and so the need for the dual tag solution is in fact lower).

I would also have minor remarks to the text:

- in page 5, line 13top - there is "at any time" - do the author mean within known/limited time?

- page 6 line 3top - there is "First." - should not it be a part of the following sentence ?

- Fig. 2 seems very dense with text hard to read - I would suggest to rearrange the elements to create more space, especially between subfigure marking ((a),(b)) and text below; a more readable font in this figure would be better too

Author Response

Thanks for providing good comments on our manuscript. Our responses are listed below, and the changed figures are in the attachment.

  1. About the main issue, there are two approaches to aligning the digital map and observed contour. One is using the two positions of two AprilTags described in our manuscript. Another is using the position and orientation of one AprilTag, provided by reviewers. In theory, their performances are the same. Thus, we modify our manuscript by adding the statement of the latter approach. However, in our experiments or some research, the orientation information provided by AprilTag is not stable. Here we provide two videos to show the stabilities using a single tag (Single Tag Video) and two tags (Two Tag Video). It can be seen that using a single tag to derive the robot's position is not stable. That is the reason we chose two tags.
    About the incorrection in Figure 3, we change the viewpoint from the digital map to the mobile robot. This is because the mobile robot detects the tags only on their front sides, even if the digital map is wrong. This change makes sense to readers.

  2. We also make changes to minor remarks:
    1. Change it to "in time" to point out that the monitoring is in quick iterations and close to real-time updating. 
    2. Replace the period with a comma.
    3. modify the figure to the proper style.

Reviewer 3 Report

The article proposes the Indoor localization method for a mobile robot using LiDAR and dual-AprilTags. The article is interesting and has practical significance. To improve the quality of the article, the following changes should be made:

  1. An answer to the question should be added to the discussion: how will the positioning accuracy change when using a large number of AprilTag (3,4,5 AprilTag)?
  2. The article should describe in more detail possible additional areas of application of the developed method, for example, in agriculture for positioning robotic platforms in rows of horticultural plantations.
  3. In the "Abstract" section, it is stated that the computer's power consumption is reduced, this issue should be considered in more detail in section "4. Experiments and results".
  4. It is necessary to consider the issue of the influence of the degree of illumination on the accuracy of positioning when using the proposed method.

Author Response

Thanks for the good comments. We made the following changes in the attachment to the comments:

  1. If a large number of AprilTags are observed simultaneously, the proposed method will process the tags pair by pair and then identify the robot position using statistical methods. Therefore, we add the part to our manuscript. The accuracy should be improved theoretically, but we do not examine the effects. 
  2. We add a paragraph in our conclusion to list possible application areas. Please see the highlights in the attachment.
  3. Section 2.3.3 describes the generation and evaluation of particles in AMCL cost computing power. Section 4.3.1, Figure 15 shows that the particles used in the proposed approach are significantly less than those used in the classic method. Therefore, we add some sentences to make the statement more clear.
  4. In the article “AprilTag: A robust and flexible visual fiducial system,” Edwin Olson describes the approach to overcome the changes in illumination. He designs the edge detector with a low-pass filter to prevent information loss due to noise sensitivity. This is the reason we did not study the illumination issue. 
